# Developmental Vitamin D Deficiency in Pregnant Rats Does Not Induce Preeclampsia

**DOI:** 10.3390/nu13124254

**Published:** 2021-11-26

**Authors:** Asad Ali, Suzanne Alexander, Pauline Ko, James S. M. Cuffe, Andrew J. O. Whitehouse, John J. McGrath, Darryl Eyles

**Affiliations:** 1Neurobiology, Queensland Brain Institute, University of Queensland, St. Lucia, QLD 4072, Australia; a.ali@uq.edu.au (A.A.); suzy.alexander@uq.edu.au (S.A.); p.ko@uq.edu.au (P.K.); j.mcgrath@uq.edu.au (J.J.M.); 2Neurobiology, Queensland Centre for Mental Health Research, Wacol, QLD 4076, Australia; 3Placental Endocrinology, School of Biomedical Sciences, University of Queensland, St. Lucia, QLD 4072, Australia; j.cuffe1@uq.edu.au; 4Autism Research Team, Telethon Kids Institute, The University of Western Australia, Crawley, WA 6009, Australia; andrew.whitehouse@uwa.edu.au; 5NCRR—National Centre for Register-Based Research, Department of Economics and Business Economics, Aarhus University, 8000 Aarhus, Denmark

**Keywords:** maternal vitamin D deficiency, preeclampsia, placental insufficiencies, renin–angiotensin system, oxidative stress

## Abstract

Preeclampsia is a pregnancy disorder characterized by hypertension. Epidemiological studies have associated preeclampsia with an increased risk of neurodevelopmental disorders in offspring, such as autism and schizophrenia. Preeclampsia has also been linked with maternal vitamin D deficiency, another candidate risk factor also associated with autism. Our laboratory has established a gestational vitamin-D-deficient rat model that shows consistent and robust behavioural phenotypes associated with autism- and schizophrenia-related animal models. Therefore, we explored here whether this model also produces preeclampsia as a possible mediator of behavioural phenotypes in offspring. We showed that gestational vitamin D deficiency was not associated with maternal blood pressure or proteinuria during late gestation. Maternal and placental angiogenic and vasculogenic factors were also not affected by a vitamin-D-deficient diet. We further showed that exposure to low vitamin D levels did not expose the placenta to oxidative stress. Overall, gestational vitamin D deficiency in our rat model was not associated with preeclampsia-related features, suggesting that well-described behavioural phenotypes in offspring born to vitamin-D-deficient rat dams are unlikely to be mediated via a preeclampsia-related mechanism.

## 1. Introduction

Emerging evidence suggests that preeclampsia is a risk factor for several neuropsychiatric disorders including autism, schizophrenia, and attention deficit hyperactivity disorder (reviewed in [1]). A recent meta-analysis showed that the risk of autism is 32% greater in children who had intrauterine exposure to preeclampsia compared with controls [2]. In this meta-analysis, seven out of ten studies reported a positive association between preeclampsia and autism.

Preeclampsia is a pregnancy-specific syndrome that affects approximately 3–7% of first pregnancies [3]. It is characterized by hypertension and renal failure [4]. Severe proteinuria after 20 weeks of pregnancy is also a common symptom [5]. About 85% of affected women present symptoms at ≥34 weeks of gestation [6]. It is one of the major causes of maternal and foetal morbidity and preterm birth [4]. The aetiology of preeclampsia has not been fully elucidated; however, clinical studies suggest that the placenta plays a central role in the development of preeclampsia [7]. Patients with preeclampsia also display a significantly altered angiogenic profile compared with normal pregnancies [4].

Preeclampsia is considered a two-stage syndrome. The first stage consists of poor placentation due to defective trophoblast invasion of the maternal spiral artery, leading to poor placental perfusion [8,9]. This leads to the second stage which proceeds with the activation of the renin–angiotensin system (RAS), placental hypoxia, placental inflammation, and increased production of soluble fms-like tyrosine kinase 1 (Flt-1) [4,10,11]. Flt-1 is a splice variant of vascular endothelial growth factor (Vegf) receptor and acts as an antagonist of Vegf and placental growth factor (Pgf) [12]. It is produced by different tissues including placenta [13]. Increased Flt-1 levels in preeclampsia have been associated with reduced free Vegf and Pgf resulting in endothelial dysfunction [10]. In addition to this, aldosterone is also suppressed in established preeclampsia [14,15]. Aldosterone is a mineralocorticoid hormone involved in the sodium reabsorption and water retention required for maternal volume expansion during pregnancy. Reduced aldosterone levels during preeclampsia may lead to reduced pregnancy-associated expansion of circulating fluid volume. This also contributes to reduced placental perfusion and ischaemia [15]. 

Clinical studies show that prenatal vitamin D (vitamin D_3_) deficiency is associated with an increased risk of preeclampsia [16,17,18,19,20]. Women with dark skin are at higher risk of vitamin D deficiency due to less efficient vitamin D synthesis [21]. The known racial disparity in preeclampsia, with women with dark skin being more likely to develop severe preeclampsia than fair-skinned women, is therefore also consistent with a possible role for vitamin D [22]. However, other studies report no evidence of an association between vitamin D deficiency and preeclampsia [23,24,25]. Furthermore, the possible mechanisms through which vitamin D deficiency can influence preeclampsia risk are unclear. Vitamin D deficiency during pregnancy also interacts with other preeclampsia risk factors such as gestational diabetes [26] and maternal obesity [27]. Vitamin D deficiency increases the risk of gestational diabetes [28]. Moreover, studies show an inverse relationship between vitamin D status during pregnancy and maternal body mass index (BMI) [29,30]. Hence, vitamin D deficiency in women with gestational diabetes and increased BMI may increase their susceptibility to preeclampsia.

Developmental vitamin D (DVD) deficiency throughout gestation is associated with the subsequent development of schizophrenia or autism in offspring [31,32,33,34]. Our laboratory has established a rat model of this exposure. We have been using this model for the past 20 years to investigate the biological plausibility of an association between gestational vitamin D deficiency and neurodevelopmental disorders such as schizophrenia and autism [35,36]. Our group has shown that offspring born to vitamin-D-deficient dams exhibit schizophrenia and autism-like behavioural phenotypes as juveniles and adults [37,38]. We have also shown a range of cellular and neurotransmitter changes in neonatal DVD-deficient brains with relevance to autism and schizophrenia [39]. However, the maternal factors that may contribute to altered foetal brain development and offspring behaviour are not fully elucidated. For instance, it is possible that the previously established phenotypes in this model may be, in part, due to impaired placental functions due to preeclampsia.

Vitamin D is a neuro-steroid which is biologically converted into its active form, 1,25-dihydroxy-vitamin D (1,25OHD), by the enzyme 1 alpha-hydroxylase. The 1,25OHD regulates expression of several genes via its nuclear receptor—the vitamin D receptor (VDR). Mounting evidence suggests that vitamin D may have a regulatory effect on the RAS system that plays a key role in regulating blood pressure. Vitamin D supplementation has been shown to normalize blood pressure in both humans [40,41] and animal models of preeclampsia [42]. It is believed that adequate vitamin D status influences blood pressure by downregulating plasma and placental Flt-1 [42]. Studies also show an inverse relationship between vitamin D and plasma renin activity [41,43], aldosterone, and angiotensin II levels [44]. Taken together, these studies suggest that vitamin D may play a role in regulating RAS to influence blood pressure. We now wish to investigate whether the behavioural phenotypes observed in DVD-deficient rats may, in part, be secondary to the impaired maternal and placental functions associated with preeclampsia.

## 2. Materials and Methods

### 2.1. Animals 

The study was approved by the Animal Ethics Committee (AEC) of the University of Queensland (Approval Number: QBI/555/16). Dietary DVD deficiency was induced according to well-established methods [45]. Briefly, standard casein rodent chow (AIN93G) free of vitamin D (Product # SF09-105) or containing 1000 IU/kg of cholecalciferol (Product # SF09-104) was fed to female Sprague–Dawley (SD) rats (Speciality Feeds, Western Australia). After 6 weeks on the diet, female rats were time-mated with vitamin-D-normal sires. Successful mating was confirmed by the presence of a copulatory plug in the vagina, and the day was denoted gestational day (GD) 0. A total of 9 control and 9 vitamin-D-deficient dams were used in this experiment. For placental analysis, two placentas per litter (placentas containing one male and one female) (control group = 9 males and 9 females, vitamin-D-deficient group = 9 males and 9 females) were used. Vitamin D deficiency was confirmed by measuring serum 25-hydroxyvitamin D_3_ (25OHD) levels at GD19 (control = 31.05 nm/L ± 8.5, deficient = 3.29 nm/L ± 1.1 (± standard deviation), (F_1,17_ = 71.30, *p* = 0.0001)).

### 2.2. Blood Pressure Measurements and Tissue Collection

Blood pressure was measured in a designated quiet room (22 ± 2 °C), where rats were acclimatized for 30 min before experiments began (Figure 1A). Blood pressure was measured by non-invasive tail-cuff plethysmography (NIBP System IN125/R ADInstruments Inc., Dunedin, New Zealand). Animals were warmed to 33 to 35 °C on a heating pad for 5 min before and during the blood pressure recordings. Rats were trained on plethysmography 2 days prior to mating. In training, rats were encouraged to walk into the rat restrainer (MLA5024, ADInstruments Inc., Dunedin, New Zealand). The pressure cuff and pulse transducer were placed at the base of the tail. To measure blood pressure, the occlusion cuff was inflated to 300 mmHg and deflated over 20 s. The volume pressure recording (VPR) sensor cuff senses variations in the tail blood volume as the blood returns to the tail during the cuff deflation. Each session involved 15 to 25 inflation and deflation cycles. The first five cycles were acclimation cycles and were not used in the analysis. Blood pressure in dams was measured at two time points. The first blood pressure reading was taken during training sessions from virgin rats and the second measurement was taken at GD18. Pregnant dams were euthanized at GD19. The dams’ blood, dams’ urine, foetal tail tips and placentas were collected. Serum was obtained from the dams’ blood. Urine was used to measure proteinuria using a Pierce™ BCA Protein Assay Kit (23227 Thermo Scientific™ Waltham, MA, USA) according to the manufacturer’s instructions. Foetal tails were used for the identification of foetal sex by amplification of the sex-determining region Y gene [46].

### 2.3. Angiotensin II, Flt-1, and Aldosterone Assays

Sera and placental samples were analysed using commercially available enzyme-linked immunosorbent assays (ELISA). Angiotensin II (ADI-900-204, Enzo Life Sciences, Inc., Farmingdale, NY, USA), Flt-1 (ab270206, abcam, Cambridge, UK) and aldosterone (ADI-900-173, Enzo Life Sciences, Inc., Farmingdale, NY, USA). ELISAs were conducted according to the manufacturer’s instructions. Briefly, serum samples were diluted (1:4) with the provided diluent and loaded onto the 96-well plates. Placental samples were prepared by homogenizing half of the placenta in lysis buffer (1:10 w/v) containing protease inhibitors. Homogenates were centrifuged at 13,000 revolutions per min for 20 min at 4 °C and supernatants collected. Total protein concentration was determined using a Pierce™ BCA Protein Assay Kit (23227, Thermo Scientific™, Waltham, MA, USA). Individual proteins were calculated relative to total protein content in homogenates. The concentrations were calculated using 4 parametric logistic regression (4PL) curve methods. 

### 2.4. Quantitative Polymerase Chain Reaction (qPCR)

A qPCR was used for the gene expression analysis of Pgf, type-1B angiotensin II receptor (Agtr1b), hypoxia-inducible factor 1-alpha (Hif1α), and prostaglandin–endoperoxide synthase 2 (Ptgs2) (see Appendix A for primer sequences). Total RNA was extracted from the remaining half of the placental tissue using an RNeasy Mini Kit (Cat No. 74104, Qiagen, Hilden, Germany) and was reverse transcribed into cDNA using a SensiFAST™ cDNA Synthesis Kit (Meridian Bioscience Inc., Cincinnati, OH, USA). The qPCR was performed using a SensiFAST™ SYBR® No-ROX Kit (Meridian Bioscience Inc., Cincinnati, OH, USA). The reaction was performed in a LightCycler^®^ 480 System (Roche Diagnostics, Penzberg, Germany) under the following conditions: denaturation at 95 °C for 5 min and then amplification for 40 cycles (95 °C for 10 s, 60 °C for 20 s, then 72 °C for 20 s). The relative gene expression was normalized to glyceraldehyde 3-phosphate dehydrogenase (Gapdh).

### 2.5. Statistical Analysis

Results were analysed using IBM SPSS (International Business Machines Corporation, Statistical Package for the Social Sciences) (IBM Corp., Armonk, NY, USA; Version 25). The blood pressure and proteinuria data in the DVD-deficient and control groups were analysed by one-way ANOVA. The Flt-1, angiotensin II, and aldosterone (ELISA) data from the dams’ sera were also analysed by one-way ANOVA. Both ELISA and RNA data from placentas were analysed by multivariant analysis of variance to measure the main effect of vitamin-D-deficient diet, main effect of foetal sex, and diet × foetal sex interactions. The placental weight was analysed by a univariant analysis of variance. Statistical significance was defined as *p* < 0.05.

## 3. Results

### 3.1. Blood Pressure and Proteinuria

Vitamin-D-deficient animals prior to mating (baseline) did not exhibit any significant (F_1,17_ = 0.02, *p* = 0.89) group difference in systolic blood pressure (Figure 1B). Blood pressure was also not different between the dietary groups at GD18 (F_1,17_ = 0.14, *p* = 0.71). There was also no difference in blood pressure between the time points tested ((baseline = 96.86 mm Hg ± 17.49, GD18 = 92.94 mm Hg ± 10.65 (± standard deviation)). There were no differences in post-pregnancy blood pressure in the two groups (F_1,17_ = 1.34, *p* = 0.26). There was also no correlation between blood pressure and observed 25OHD concentration in either vitamin-D-deficient or control dams (see pairwise correlations between vitamin D levels and blood pressure in Appendix A). Moreover, no difference was observed in total protein concentrators in urine collected from vitamin-D-deficient and control groups at GD18 (F_1,17_ = 0.77, *p* = 0.39) (Figure 1C). 

### 3.2. Angiotensin II, Flt-1, and Aldosterone Levels in Dam Sera

Pregnant animals at GD19 displayed no significant difference (F_1,17_ = 2.37, *p* = 0.14) in the circulatory levels of Flt-1 between control and vitamin-D-deficient groups (Figure 1D). There was also no significant effect of diet on angiotensin II (F_1,17_ = 0.66, *p* = 0.43) (Figure 1E) and aldosterone protein levels (F_1,17_ = 1.09, *p* = 0.75) in the dams’ sera (Figure 1F).

### 3.3. Placental Weight and Angiotensin II, Flt-1, and Aldosterone Levels in Placenta

There was no main effect of diet (F_1,35_ = 1.05, *p* = 0.31) or main effect of embryo sex (F_1,35_ = 2.30, *p* = 0.14) on placental weight. Nor there was any diet × embryo sex interaction (F_1,35_ = 1.57, *p* = 0.22) on the weight of placentas.

Consistent with the results from the dams’ sera, there was no main effect of diet on Flt-1 (F_1,35_ = 1.37, *p* = 0.25), angiotensin II (F_1,35_ = 0.51, *p*  = 0.82), or aldosterone (F_1,35_ = 1.50, *p* = 0.23) in placenta. Furthermore, no effect of foetal sex was found on Flt-1 (F_1,35_ = 0.40, *p* = 0.53), angiotensin II (F_1,35_ = 0.35, *p* = 0.56), or aldosterone (F_1,35_ = 0.04, *p* = 0.94) in placenta. No diet × foetal sex interactions were observed for any of these hormones (Figure 2).

### 3.4. mRNA Expression in Placenta

Maternal vitamin D deficiency has no effect on gene expression of *Agtrb1* (F_1,35_ = 0.53, *p* = 0.82), *Pgf* (F_1,35_ = 0.40, *p* = 0.53), *Hif1a* (F_1,35_ = 0.90, *p* = 0.35), or *Ptgs2* (F_1,35_ = 0.09, *p* = 0.93) (Figure 3) in placenta. There was also no main effect of foetal sex on expression of any of the genes *Agtrb1* (F_1,35_ = 1.46, *p* = 0.23), *Pgf* (F_1,35_ = 0.83, *p* = 0.77), *Hif1a* (F_1,35_ = 0.11, *p* = 0.74), or *Ptgs2* (F_1,35_ = 1.02, *p* = 0.31). No diet × sex interactions were found (Figure 3).

## 4. Discussion

The primary aim of this study was to analyse whether vitamin D deficiency induces preeclampsia-like features in vitamin-D-deficient pregnant rats. We showed that vitamin D deficiency was not associated with maternal hypertension or proteinuria. Moreover, maternal and placental angiogenic factors were not significantly altered by maternal vitamin D deficiency.

Despite the fact that the epidemiological evidence linking vitamin D deficiency in pregnancy and preeclampsia is reasonably strong [16,17,18,19,20], the molecular basis for dietary vitamin D deficiency inducing preeclampsia is less well established. The molecular evidence for the link between vitamin D deficiency and preeclampsia has primarily been established from studies that used constitutive mouse knock-outs of either the VDR or the enzyme 1 alpha-hydroxylase, which is responsible for the production of the active vitamin D hormone [47,48,49]. Ablation of either VDR or 1 alpha-hydroxylase in mice activates the RAS and leads to the accumulation of angiotensin II [47,48,49]. The deletion of 1 alpha-hydroxylase also leads to hypertension and cardiac hypertrophy in mice. Interestingly, vitamin D supplementation normalizes the blood pressure and RAS in 1 alpha-hydroxylase knockout mice, suggesting that vitamin D may be protective against preeclampsia [49]. The situation with respect to dietary deficiency, however, is far less clear. When *dietary* vitamin D deficiency is induced in pregnant mice, enhanced blood pressure accompanied by elevated RAS, *Agrtb1*, and dysregulated placental vascularization was observed [50]. However, in contrast, Andersen et al. showed no effect of dietary vitamin D deficiency on key aspects of preeclampsia phenotype in a transgenic rat model of human renin–angiotensin system-mediated preeclampsia [51].

In considering these inconsistencies in studies of dietary vitamin D deficiency, it is firstly important to consider the differences between rat and mouse placentas. During rat pregnancy, placental trophoblast cells enter the uterine decidua and invade the maternal endometrium; however, this trophoblast invasion does not extend into the myometrium in mice [52,53,54]. Given that a deeper trophoblast invasion in rat is likely to allow greater placental perfusion [55], this may indicate that the rat placenta is more resilient to preeclampsia compared to the mouse placenta. Alternatively, differences between species in corticosterone production in response to dietary vitamin D deficiency during pregnancy may also be important to consider. Elevated levels of cortisol during pregnancy induce hypertension and endothelial dysfunction in women [56,57]. It is interesting to note that mice and rats show differential corticosterone responses to gestational vitamin D deficiency. Gestational vitamin D deficiency in mice leads to increased maternal corticosterone levels and down-regulation of placental enzymes which inactivate maternal corticosterone [58]. However, we have consistently observed no effect of vitamin D deficiency on baseline corticosterone levels in pregnant rats [59,60]. Thus, differences between placental architecture and/or hypothalamic–pituitary–adrenocortical axis activation may explain the absence of preeclampsia phenotypes between species when subjected to gestational vitamin D deficiency.

Inconsistencies between species were also observed in a widely used reduced uterine perfusion pressure (RUPP) model of preeclampsia, induced by surgical restriction of blood flow to the ovarian arteries [61]. In contrast to RUPP mice, RUPP rats show more severe features of preeclampsia. For example, proteinuria and cytokines, and Vegf levels, were elevated in RUPP rats [62,63] but not in RUPP mice [64]. Therefore, the contribution of these pathways in modifying preeclampsia features in both species is an important topic of future research. Differences in the gestational length and time of the RUPP procedure may also contribute to the severity of preeclampsia features between species.

Our findings lend no weight to the hypothesis that molecular alterations in RAS factors and placental inflammation may account for the clinical observations linking vitamin D deficiency and an increased risk of preeclampsia. When preeclampsia proceeds to its second phase after poor placentation and uneven blood perfusion, hypoxia and inflammation generally follow. This manifests in clinical preeclampsia in humans [4] and leads to the activation of RAS and the placental inflammatory mediators *Hif1a* and *Ptgs2* [65,66,67]. Several ex vivo studies have shown that vitamin D suppresses *angiotensin II*, *Hif1a*, and *Ptgs2* in different cell lines [68,69,70,71]. The absence of such alterations also weakens the hypothesis that these mechanisms underpin the clinical association between vitamin D deficiency and preeclampsia. 

This study has some limitations. Many studies in animal models of preeclampsia examine longitudinal changes in blood pressure from mid to late gestation [50,72]. However, we measured blood pressure only at a single time point during pregnancy; thus, we may have missed any longitudinal changes associated with preeclampsia. In addition, we used an indirect method (tail-cuff sphygmomanometer) of measuring systolic blood pressure in vitamin-D-deficient rats. Furthermore, the requirement for restraint, room temperature fluctuations, animal handling, and equipment calibration may produce artefacts [73,74].

## 5. Conclusions

Gestational vitamin D deficiency has been implicated as a risk factor for neuropsychiatric disorders such as autism and schizophrenia [31,32,33,34]. Vitamin D deficiency in pregnancy has also been linked with preeclampsia in clinical studies and, in turn, preeclampsia has also been epidemiologically linked with these psychiatric disorders [16,17,18,19,20]. The DVD-deficient rat is becoming a more widely used model to investigate the neurobiology of later psychiatric disorders. The current study was designed to investigate whether preeclampsia-like phenotypes contribute to this association. This study showed no association between dietary vitamin D deficiency and preeclampsia-like phenotypes or molecular correlates. We conclude that preeclampsia in the pregnant vitamin-D-deficient rat is not a contributing factor to the well-described behavioural phenotypes of relevance to autism and schizophrenia seen in this model [36,75]. This also suggests that the gestational vitamin-D-deficient rat model may not be a good model for examining maternal factors associated with preeclampsia.

## Figures and Tables

**Figure 1 nutrients-13-04254-f001:**
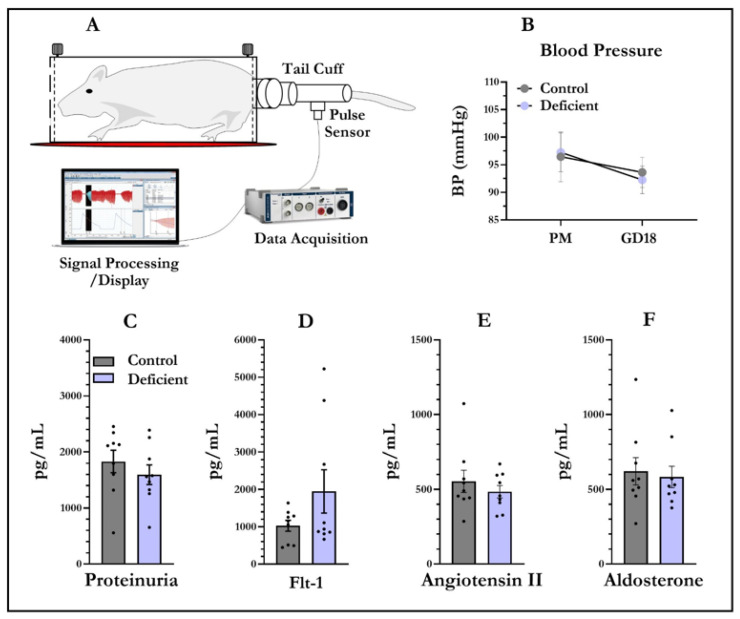
Vitamin D-deficient pregnant rats do not display preeclampsia phenotypes. Blood pressure data were acquired and processed using LabChart^®^ 8.1 (**A**). Blood pressure was recorded from virgin and pregnant rats (**B**). There was no significant effect of diet on blood pressure at both time points. No proteinuria difference was detected in vitamin-D-deficient dams (**C**). Maternal Flt-1 (**D**), angiotensin II (**E**) and aldosterone (**F**) were also not significantly changed between the sera of control and vitamin-D-deficient dams. Data shown are means; error bars show SEM; *n* = 9 control, *n* = 9 vitamin-D-deficient, PM = prior to mating, GD18 = gestational day 18.

**Figure 2 nutrients-13-04254-f002:**
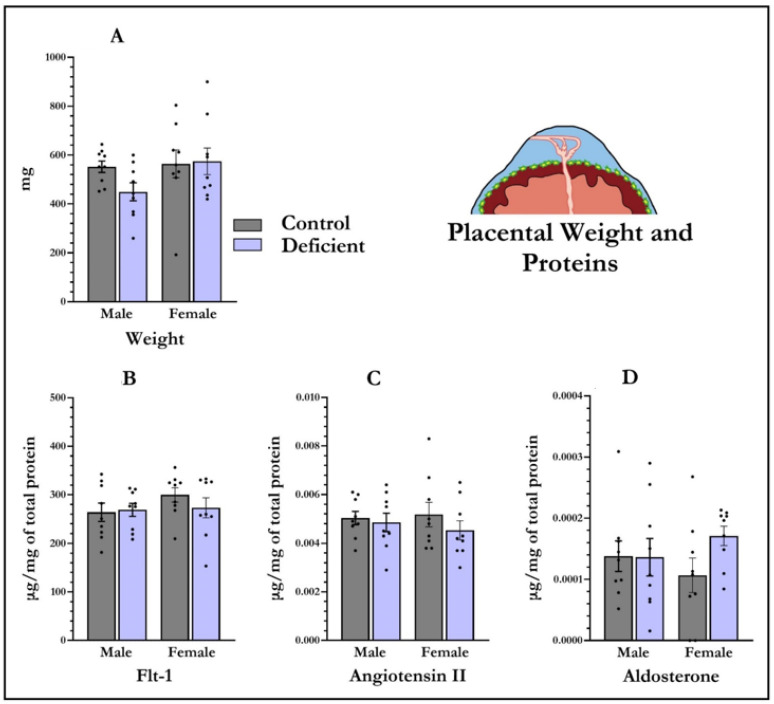
Vitamin-D-deficient pregnant rats do not have alterations in placental weight or expression of proteins central to preeclampsia phenotypes. Placental weight and placental proteins were quantified at gestational day 19. There was no effect of diet on placental weight (**A**). There was also no effect of diet or embryo sex on placental Flt-1 (**B**), angiotensin II (**C**), or aldosterone (**D**). Data shown are means; error bars show SEM; *n* = 9 control males, *n* = 9 vitamin-D-deficient males, *n* = 9 control females, *n* = 9 vitamin-D-deficient females.

**Figure 3 nutrients-13-04254-f003:**
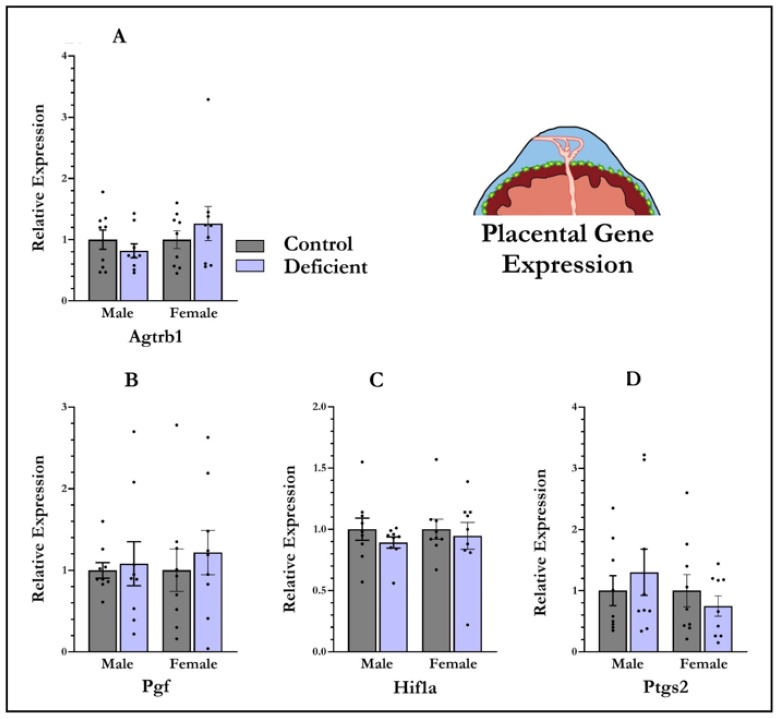
Vitamin-D-deficient pregnant rats do not have alterations in expression of genes central to preeclampsia phenotypes. The mRNA levels were measured at gestational day 19 from placentas collected from both male and female foetuses. No significant effect of diet was observed on type-1 angiotensin II receptor (*Agtrb1*) (**A**), placental growth factor (*Pgf*) (**B**), hypoxia inducible factor 1 alpha (*Hif1a*) (**C**), or prostaglandin–endoperoxide synthase 2 (*Ptgs2*) (**D**). Data shown are means; error bars show SEM; *n* = 9 control males, *n* = 9 vitamin-D-deficient males, *n* = 9 control females, *n* = 9 vitamin-D-deficient females.

## Data Availability

The data presented in this study are available on request from the corresponding author.

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
