# Peer review of "Developmental Vitamin D Deficiency in Pregnant Rats Does Not Induce Preeclampsia"

_nutrients, 2021, doi:10.3390/nu13124254_

Round 1
Reviewer 1 Report
Asad Ali et al studied the effects of vitamin d-depleted diet on Sprague-Dawley rat pregnancies with the hypothesis that vitamin D depletion could induce preeclampsia-like features. The manuscript is well written. The paper shows all negative data, which ultimately makes it less novel and interesting, and their model and methods has multiple limitations limiting the enthusiasm of the reader. The discussion needs significant improvement as noted in the major critiques.
Major critiques:
- The most important flaw of this model is that Sprague-Dawley rats are a very healthy and robust rat strain. Inducing preeclampsia-features in this rat strain is not an easy task. Therefore, it was to be expected that negative data would ensue. Vitamin D deficiency studies are always best in a model that already shows preeclampsia-features. Currently, it is thought that vitamin D deficiency contributes rates than ‘causes’ preeclampsia. And some women are likely to be more at risk of preeclampsia if they are VDD than other women. VDD interacts with obesity and other PE risk factors. Altogether, it is truthfully too minimalistic to explore VDD as a cause of PE. The discussion needs to address all of these features before this study is published.
- Further limitations of this model, is the comparison of the Sprague-Dawley rat with any mouse model. Mice have a different vitamin D metabolism and this species has higher predisposition for preeclampsia induction compared to the rat (please check multiple reviews on animal models of preeclampsia). In order of susceptibility is guinea pig>mice>rat. This is shown very nicely with the RUPP model where mice have more severe features of PE compared to rats with the same surgical procedure. This could explain why VDD mice had preeclampsia features (your reference number 46) and your rat model didn’t. I see the authors try to explain the differences between VDD mice/rats in lines 244-260. However, this section should be expanded.
- Moreover, knockout models (VDR null, CYP27B1 null) cannot be compared to the VDD model. The levels of 25-OH-D are extremely low, but they are still sufficient to provide the normal levels of 1,25-OH2-D (normal serum range between ~25-75 pg/mL that can be derived from 3 ng/mL in VDD rats). That is why the model is not as clean as a knockout model, in addition to SD rats having a very robust and healthy pregnancy outcomes. Perhaps using other rat strains like DSS, BN, or the SHR rat strains, the VDD depletion would lead to preeclampsia symptoms.
- The use of tail-cuff plethysmography and Pierce BCA assay for proteinuria are not the most sensitive methods. However, for this study, the methods used in this study are sufficient to show the lack of PE features. But it should be noted under their limitations paragraph. Another limitation is that they only studied BP and proteinuria at GD18-19 and at pre-pregnancy. This should be clearly noted as a limitation. BP and proteinuria could rise shortly with later resolution.
- In the discussion, the authors need to clarify the comparison with other models of preeclampsia, more than with knockout models, unless the data shown from the knockout models are specific to pregnancy (lines 233-237).
- An entire paragraph in the discussion needs to address the strengths and limitations of this study, accordingly to what is written above. The most crucial aspect is that the authors cannot decidedly conclude that VDD does not contribute to preeclampsia when studying a very healthy rat strain. This is also shown in women without risk factors for PE can have VDD without any obstetric complication, even with obesity.
Minor critiques:
- Introduction: need to include the proposed role of VDD on preeclampsia, not just epidemiological data. For instance: show interaction of VDD with other PE-risk factors.
- Figure 1 (d, e, and f): indicate if these assays are in maternal sera
- Line 267: delete the word ‘angiogenesis’ and write ‘angiotensin’
Reviewer 2 Report
Major comments:
- Please be accurate throughout the whole manuscript which vitamin D metabolite is meant when the term "vitamin D" is used; vitamin D3, 25(OH)D3 or 1,25(OH)2D3?
- The main problem of this study is that it ends with negative results. Is there now way to extract other "positive" results from the study. Thus, which effects of vitamin D could be observed, if the effect of preclampsia is negative
Minor comments:
- Please better avoid DVD as abbreviation for developmental vitamin D.
- Please use latest nomenclature for gene and protein names.
- Please better use A, B, C... in the figure legends instead of Fig. 1A, Fig. 1B, Fig. 1C... for all figures.
- All gene and mRNA gene abbreviations should be in italic.
- 5. Pease check for typos, e.g. i line 274 one "e" too much.
Round 2
Reviewer 2 Report
All figures still contain a "Fig X" type of heading, which better should be reduced to the subfigure caption, such as A, B, C....